# A workplace organisational intervention to improve hospital nurses' and physicians' mental health: study protocol for the Magnet4Europe wait list cluster randomised controlled trial

Walter Sermeus [1], Linda H Aiken,[2] Jane Ball,[3] Jackie Bridges [3], Luk Bruyneel,[1] Reinhard Busse,[4] Hans De Witte,[5,6] Simon Dello [1], Jonathan Drennan,[7] Lars E Eriksson [8], Peter Griffiths [3], Dorothea Kohnen [5], Julia Köppen,[4] Rikard Lindqvist,[8] Claudia Bettina Maier,[4] Matthew D McHugh [2], Martin McKee [9], Anne Marie Rafferty,[10] Wilmar B Schaufeli,[5,11] Douglas M Sloane,[2] Lisa Smeds Alenius,[12] Herbert Smith,[2] Magnet4Europe consortium

For numbered affiliations see end of article.

**Correspondence to**
Dr Walter Sermeus;
walter.sermeus@kuleuven.be

## ABSTRACT

**Introduction** The increasing burden of mental distress reported by healthcare professionals is a matter of serious concern and there is a growing recognition of the role of the workplace in creating this problem. Magnet hospitals, a model shown to attract and retain staff in US research, creates positive work environments that aim to support the well-being of healthcare professionals.

**Methods and analysis** Magnet4Europe is a cluster randomised controlled trial, with wait list controls, designed to evaluate the effects of organisational redesign, based on the Magnet model, on nurses' and physicians' well-being in general acute care hospitals, using a multicomponent implementation strategy. The study will be conducted in more than 60 general acute care hospitals in Belgium, England, Germany, Ireland, Norway and Sweden. The primary outcome is burnout among nurses and physicians, assessed in longitudinal surveys of nurses and physicians at participating hospitals. Additional data will be collected from them on perceived work environments, patient safety and patient quality of care and will be triangulated with data from medical records, including case mix-adjusted in-hospital mortality. The process of implementation will be evaluated using qualitative data from focus group and key informant interviews.

**Ethics and dissemination** This study was approved by the Ethics Committee Research UZ/KU Leuven, Belgium; additionally, ethics approval is obtained in all other participating countries either through a central or decentral authority. Findings will be disseminated at conferences, through peer-reviewed manuscripts and via social media.

**Trial registration number** ISRCTN10196901.

## INTRODUCTION

Mental health and well-being are high priorities for the European Union (EU).[1] Mental health conditions account for 22% of the

## STRENGTHS AND LIMITATIONS OF THIS STUDY

⇒ Magnet4Europe uses a cluster randomised controlled trial design with wait list controls in over 60 hospitals in six European countries.
⇒ A multicomponent strategy is used to support implementation of a complex intervention at the hospital level, aimed to improve clinician well-being.
⇒ The Magnet4Europe intervention is tailored to European hospitals, facilitating implementation and transferability without disrupting clinical practice.
⇒ Contextual factors such as different health systems within and across countries may impact the standardised Magnet4Europe intervention and unintentionally lead to variation in the intervention and outcomes.
⇒ Embedding a complex intervention at hospital-level takes time and any effect might become apparent only after the study ends.

EU's disease burden measured in Years Lived with Disability, thus imposing a significant burden on individuals, society and the economy.[2] Health workers experience greater levels of job-related burnout and other mental health disorders than those in other sectors.[3,4] An earlier European study of working conditions in European hospitals, identified high rates of job dissatisfaction and burnout, with burnout rates among nurses varying from 10% to 78%[5,6] while the corresponding figures from studies of physicians ranged from 25% to 60%, varying among organisations and medical specialties.[7–10] Burnout in the healthcare workforce not only

impacts on those experiencing it, leading to depression, substance abuse and even suicide[11–15] but is also associated with worse patient outcomes, lower patient satisfaction,[6] medical errors,[16] reduced quality and safety[17] and reduced efficiency of hospitals.[18] The resulting lost productivity, combined with current recruitment challenges, further threatens the already overstretched health workforce, widening the gap between provision of health services and population needs.[19]

This situation has been exacerbated by the COVID-19 pandemic,[20] increasing anxiety and stress among clinicians[21] and further impacting on health workers' mental health.[22–24]

Effective, affordable and sustainable interventions to improve health professionals' mental health and well-being are essential if we are to interrupt the vicious cycle of high burnout, worsening mental health, lost productivity and unsafe care.[25] Interventions to prevent burnout can be divided into two types: first, psychological interventions aimed at individual level and second workplace redesign. There is only limited evidence of the impact of individual psychological approaches to improve resilience and coping skills (eg, mindfulness).[26] The second approach, redesigning the organisational environment in which health professionals provide care, aims to reduce modifiable sources of clinician stress and burnout, reinforce clinicians' perceptions of well-being and enhance their autonomy and control over the conditions of their work. There is substantial research pointing to organisational redesign as the most promising of these two approaches.[27]

The finding that the causes underlying burnout in nurses and physicians are similar and, especially, are rooted within the work environment[28–31] provides a compelling rationale for prioritising the redesign of hospital environments to address this issue. In the 1980s, hospitals in the USA were faced with high nurse turnover,[32 33] low retention, and increasing early retirement rates.[34] Despite the general nursing shortage, some US hospitals—later referred to as Magnet hospitals—were more successful in recruiting and retaining nurses by creating a positive work environment. Features characterising these hospitals included flat decentralised organisational structures, empowering frontline staff in decision-making and transformational leadership.[35] The original characteristics of these hospitals[36] were developed into 14 forces of Magnetism and later configured into 5 components of the Magnet model that was used to establish the criteria for Magnet designation.[37] A robust body of evidence, primarily stemming from US hospitals, documents the association of the Magnet model with improved well-being of staff, including lower burnout, higher job satisfaction, lower intent to leave their job,[38 39] in addition to positive financial outcomes for organisations,[40 41] higher levels of patient satisfaction[42 43] and improved clinical outcomes.[44 45] Longitudinal panel studies have found that hospitals that follow the Magnet model improve their work environments, well-being of staff and patient

outcomes, compared to other hospitals. Two international pilots tested the transferability of the Magnet model, demonstrating the feasibility of achieving positive results outside the USA. One, in England, focused primarily on the implementation of the Magnet process.[46] The other, a four-hospital intervention with accompanying evaluation in Russia and Armenia, used the Magnet model with twinning with Magnet recognised hospitals.[47] In both cases, the intervention hospitals improved their work environments substantially from their baselines within the 2-year intervention and achieved better outcomes for staff, such as reduced dissatisfaction, intent to leave and emotional exhaustion (EE), while perceived quality of care improved. Despite the large body of evidence and recent uptake in two European hospitals, Magnet organisational redesign initiatives have not been adopted widely in Europe.

The aim of the Magnet4Europe project is to evaluate the ability of organisational redesign in European hospitals to improve nurses' and physicians' mental health and well-being. The organisational redesign intended in Magnet4Europe is a system-level intervention, targeting the hospital as a whole on the level of the organisation. Through co-creation and stakeholder co-designed adaptations, the Magnet model is adapted to the European context, providing the first element and basis for the Magnet4Europe intervention. In addition to this first component, one-to-one twinning with Magnet recognised hospitals, learning collaboratives and a critical mass of hospitals are also introduced into the intervention. While the Magnet4Europe intervention shares common elements with other organisational interventions, the combination of all the above components, targeting two major clinical professions—both nurses and physicians—and its primary focus on the work environment as levers for change differentiate the Magnet4Europe intervention from any other organisational intervention and make it unique in its kind.[48]

## METHODS AND ANALYSIS
### Study design and setting
Magnet4Europe will use a usual-practice cluster randomised controlled trial with wait list controls in a nested mixed-methods evaluation. The study protocol for this study follows 'the Standard protocol items: recommendations for intervention trials 2013 statement' (SPIRIT, online supplemental file 1).[49] The setting will be acute general hospitals in six European countries, including both EU member states (Belgium, Germany, Ireland and Sweden) and non-EU member states (England and Norway). These six countries also represent two dominant types of health systems, ie, Bismarck and Beveridge,[50] that vary with regards to the extent of national government intervention in health services delivery, something that may be a factor in uptake of innovation.

## Eligibility

European hospitals are eligible to participate if: (1) no Magnet designation by the American Nursing Credentialing Centre (ANCC) had been acquired in the past or at the time of the start of the intervention, (2) bed size ≥150 and (3) the hospital is focused on acute care for adults, including wards in the area of internal medicine and/or surgery. Excluded are highly specialised hospitals, for example, psychiatric hospitals, tropical medicine or paediatrics.

Within each hospital, registered nurses and physicians (including residents) will be eligible to participate in the quantitative and qualitative evaluation of the intervention if they: (1) have direct patient contact, (2) meet the minimum qualifications as stipulated by Directive 2013/55/EU amending Directive 2005/36/EC on the recognition of professional qualifications and (3) work on adult inpatient units including intensive care units and the emergency room. Nurses and physicians working in specialised units such as neonatology, paediatrics, obstetrics, psychiatry, operating room, pathology, microbiology, radiology and medical imaging will be excluded.

## Study sample

A two-phased non-probability strategy will be used to sample organisations at the hospital level. First, in February and March 2019, principal investigators in the six European countries contacted eligible hospitals (see below) within those countries to consider taking part in the study using various recruitment strategies. Hospitals were informed of the Magnet4Europe objectives and methodology and were invited to show their interest in participating by submitting a letter of intent by March 2019. In spring 2020, participating hospitals (some of the original expression of interest sites and other new ones) were asked to confirm their commitment by signing an agreement committing to participate for the full duration of the study period and adhere to the Magnet4Europe protocol.

Sample size will be determined based on the power calculation for a cluster randomised controlled trial with before-and-after measures.[51] The calculated design effect is based on average cluster size and the intraclass correlation coefficient (ICC), a measure of the amount of dependency among observations within classes (ie, hospitals in our study).[51 52] Power calculation will be performed for the primary outcome of interest, burnout, measured by the EE subscale of the Maslach Burnout Inventory (MBI).[53] The estimated prevalence rate of EE is 30% based on the team's earlier RN4CAST study.[6] In order to detect a 20% relative reduction (ie, a 6 percentage point reduction) in burnout with a power of 80% and an alpha of 5%, an average cluster size of 200 and an ICC of 0.025,[54] this study will require 10 253 health professionals in 51 clusters per measurement occasion (at endpoint of the study).

Based on the power calculation, we aim to include 60 general acute care hospitals in six countries (Belgium, England, Germany, Ireland, Norway and Sweden) and recruit at least 200 professionals per hospital.

## Randomisation

Randomisation to intervention and control group is at the hospital level. Within each country, hospitals will be matched on three characteristics, (1) bed size, (2) teaching status and (3) high technology, using a Euclidian distance matrix based on the smallest within pair covariate distance. Subsequently, simple random sampling—allowing for selection with equal probability—will be used to allocate one hospital within each matched pair of hospitals to the immediate intervention group or the wait list control group. Both immediate intervention group and wait list controls will be informed which group they have been allocated to. Given the characteristics of the intervention in which active participation of the hospitals is required, blinding to the intervention will not be possible.

## Intervention

A multicomponent intervention comprising organisational redesign of hospitals will be implemented in both arms of the study. The intervention was launched in the immediate intervention group in month 10 of the project (October 2020) and will have a total duration of 31 months. The intervention in the usual-practice wait list control group will be initiated in month 17 (May 2021) and will last for 24 months (figure 1).

The intervention consists of five distinct components: The Magnet blueprint, one-to-one twinning between European hospitals and US Magnet designated hospitals, learning collaboratives involving group meetings to share best practices, critical mass and communication and feedback.

The first part of the intervention comprises the elements outlined in the Magnet manual of organisational redesign. This contains a step-by-step blueprint through the five overarching Magnet components: (1) structural empowerment of clinical staff, (2) transformational leadership, (3) exemplary and evidence-based professional practice, (4) new knowledge, innovations and improvements and (5) empirical outcomes. These principles serve as continuous feedback loops as to whether organisational changes are producing the intended outcomes for health professionals and patients. The manual provides definitions of the principles and gives examples of evidence-based indicators that reflect progress towards achieving them. Using the Magnet4Europe Gap Analysis Tool (adapted from the ANCC Magnet Gap Analysis tool), each participating European hospital will, in collaboration with their Magnet twinning partner, perform a gap analysis that will illuminate the gap between the as-is situation and the aspirational organisational features as delineated in the Magnet blueprint. The tool should be used for initial and continuous assessment of workplace redesign. This will increase the ability of hospitals to identify and prioritise learning and infrastructure needs, while also identifying

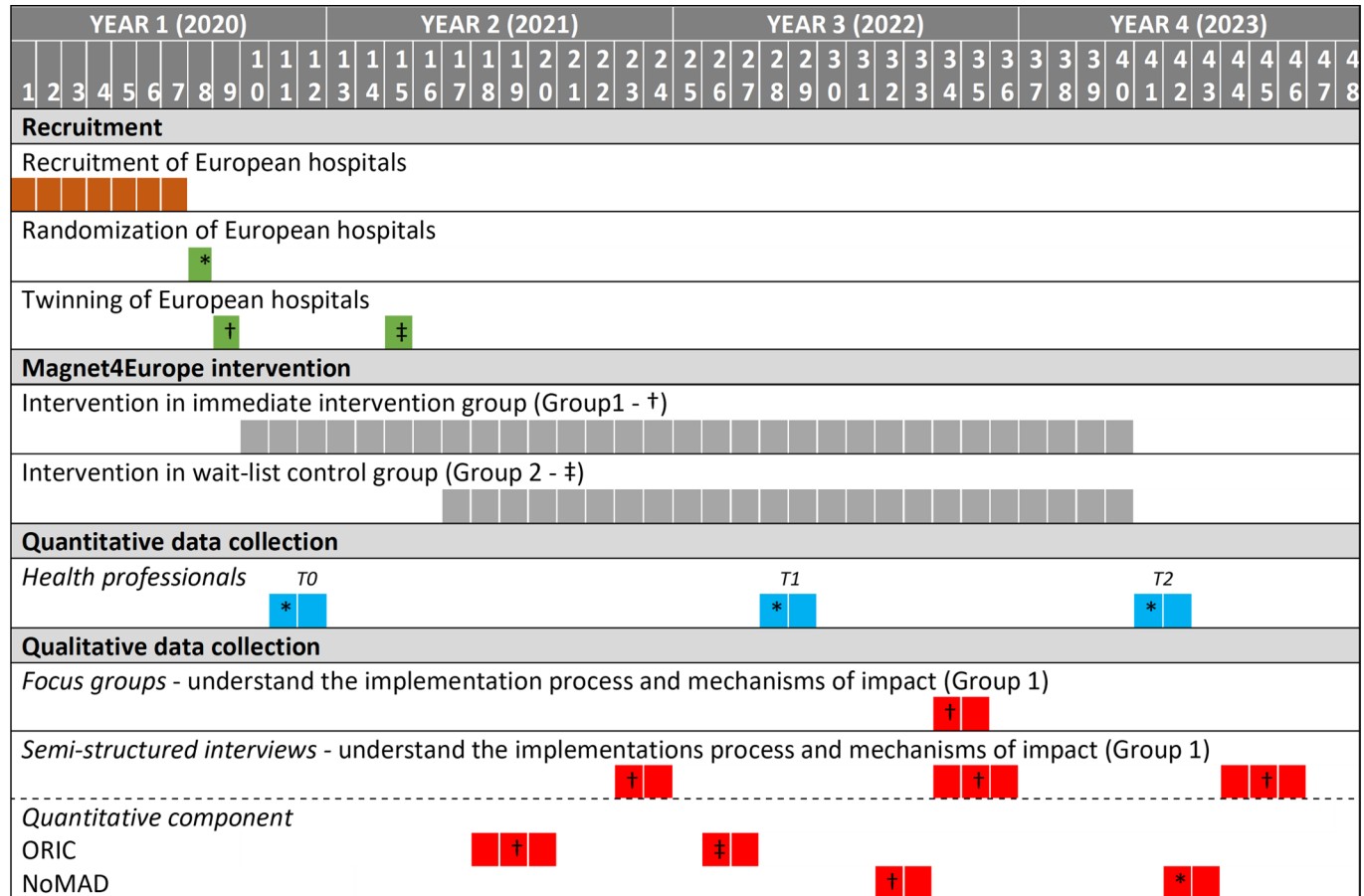

**Figure 1** Magnet4Europe timeline. †Activity for group 1; ‡Activity for group 2; *Activity for both groups.

initiative-wide gaps and exemplars for systematic improvement and recognition. Drawing on the results of the gap analysis, each hospital will be responsible for tailoring and individualising the interventions to their hospital-specific context and developing a concrete, written action plan that will be monitored and executed during the course of the intervention period. Results of the gap analysis and the effect of tailored interventions will serve as feedback to enhance learning.

The second intervention component consists of one-to-one twinning between European hospitals aspiring to implement the Magnet model and longstanding US Magnet designated hospitals. This seeks to promote capacity building through transfer of knowledge, skills, tools, technologies and best practices, thereby supporting positive reciprocal organisational changes.[55] An equal number of US Magnet hospitals—recognised for having an excellent work environment—will be recruited; allowing for a 1:1 twinning ratio of European intervention hospitals with US Magnet Hospitals. The twinning pairs will meet every two weeks virtually and two face-to-face mentoring sessions are planned each year when travel is safe. Face-to-face meetings between both parties allow for a profound, impact- and meaningful experience that maximises the effect of the capacity building activities facilitating the intended organizational redesign.

Third, learning collaboratives will take place throughout the duration of the project and will be accessed and attended by all European hospitals in which the intervention is taking place at that point in time. The learning collaboratives will also involve policymakers, so as to promote sustainability. Learning collaboratives build on the concept of communities of practice,[56] involving people coming together to engage in a process of learning in a shared domain of interest. These will be held monthly using remote means of communication with the possibility of in-person meetings annually when travel is safe. Effective elements of collaboratives, that is, learning sessions, action periods and a collaborative extranet, will be implemented.[57–59]

Fourth is the creation of a critical mass of participating hospitals promoting innovation, attracting public interest, engaging stakeholders and fostering replication and scaling of the intervention. Achieving a critical mass is essential for sustainability and scale-up of the intervention.[60] Critical mass shifts the context from one in which the change activities are unusual to one in which they are the norm. This long-standing principle in organisational sociology suggests that organisations become more open to and actively seek out change as the intervention gains legitimacy among peer institutions; this is particularly true in complex workplaces such as those found in

healthcare.[61 62] Critical mass also maximises the network exchanges and opportunities for feedback that arise out of other elements of our intervention such as the learning collaborative. This is supported by recent experience with the Magnet model.[63 64]

A fifth component of the intervention is to provide regular survey feedback to hospitals on clinicians' reports on work conditions and well-being on the hospital level, separately for physicians and nurses.[65] Hospitals will be anonymised using a randomly generated letter code allowing each hospital to identify themselves but not any of the other hospitals. Data will only be reported in aggregated form and will not highlight a single individual's data.

## Evaluation of the effect of the intervention

A mixed-methods evaluation will be conducted to examine the effect of the intervention, including quantitative and qualitative components.

The guiding framework for our quantitative evaluation is the Quality Health Outcomes Model—as adapted by Aiken and colleagues[66]—based on a conceptual framework that sets out mechanisms relating organisational attributes to clinician and patient outcomes. Their model is theoretically founded on Donabedian's early conception of associations between structures, processes and outcomes[67] including the dynamism between model components that was proposed in the Health Quality Outcomes Model.[68] The Job Demands-Resources model also informed the choice of survey items, seeking to understand the intervention's impact on the imbalance between demands on the clinician and the resources available to clinicians to deal with these demands.[69] Job demands refer to aspects of the job that require sustained physical or mental effort and are associated with certain physiological and psychological costs. Job resources describe aspects of the job that are related to achieving work goals, reducing job demands and the associated physiological and psychological costs and stimulating personal growth and development.

The qualitative part of the study aims first, to evaluate the implementation process, adaptation, reach and quality as well as mechanisms of impact manifested in participants' responses, mediators, unexpected pathways and consequences; and second, to explore barriers and enablers of the implementation and the experiences of the change process from the perspectives of health professionals and persons involved in leading the implementation.[70]

## Quantitative evaluation
### Quantitative data collection

Quantitative data collection across countries is standardised using an online data collection platform. Quantitative survey data will be collected at three points in time, that is, at baseline (T0) in months 11–12 of the project timeline, at T1 (months 28–29) and T2 (months 41–42) (figure 1) during an 8week period. An open cohort design will be employed in which all eligible nurses and

physicians are invited to participate in each of these three measurement points (T0–T2). Nurses and physicians will be recruited within each hospital at baseline and will be asked to participate in surveys with repeated measures taken over the three points in time allowing partially longitudinal follow-up of a cohort of individuals. Initial non-participants and new employees will be invited to join the study at later measurement points. Regular reminders will be sent to any clinicians (registered on the platform) through the online data collection platform who have not or only partially completed the survey. Individual informed consent for participation will be obtained by the researchers at the first contact (online supplemental file 2). Each record of informed consent is retained on the online platform.

### Quantitative measures

The main survey for nurses and physicians contains a core set of outcomes and main explanatory and process variables directly linked to the Magnet4Europe Magnet Gap Analysis Tool. All study measures are based on validated instruments.

All survey items will be translated in Dutch, English, German, Norwegian and Swedish. When available, validated translations of instruments will be used. If no validated translation is available, a prespecified translation procedure will be applied. First, a forward translation will be performed by a researcher involved in this research; the language in which the item is translated is the native language of the researcher. Second, other researchers within the country research group will thoroughly assess the forward translation in multiple phases until consensus is reached on each item.

### Primary outcome

Burnout is the primary outcome measure and will be measured on individual level using the 9-item EE subscale of the MBI.[53] In addition, the Burnout Assessment Tool (BAT), a 12-item novel self-report questionnaire will also be used to measure burnout.[71–73] The combination of both instruments to assess burnout, allows for: (1) a direct comparison of levels of EE with previous studies in the medical field (using the EE-subscale of MBI) and (2) an up-to-date assessment of the more comprehensive burnout syndrome as well as a reliable estimation of the number of respondents with severe burnout symptoms and a comparison with the level of burnout of the national workforce.

### Covariates

Secondary outcome measures include work engagement (UWES-3),[74] job satisfaction,[75 76] depression (PHQ-2),[77] anxiety (GAD-2),[78] general health (SF-8),[79] sleep quality (PSQ),[80] intent to leave the hospital,[75 76] absenteeism and presenteeism (HPQ),[14] workability, work-life conflict, team commitment, organisational commitment,[81] and whether clinicians would recommend the hospital.

Core exploratory variables include work environment (PES-NWI),[82] various measures of staffing and workload (NPQS),[83] care left undone,[84] operational failures, emotional demands (SIMPH),[85] red tape,[86] role conflicts (NPQS), job control, skill use, value congruence, performance feedback, opportunities for learning and development[87] and intrinsic motivation (WEIMS).[88]

Supplementary survey items are used to test more specific hypotheses.[69 89] These items contain measures on qualitative job insecurity,[90] emotional dissonance,[91] task variety (QEEW),[92] role clarity (NQPS),[83] basic need satisfaction[93] and engaging leadership.[94]

Productivity measures consist of staff turnover rate, sickness absence rate, number of vacancies and the use of agency staff—measured at organisation level—complemented by previously mentioned secondary outcome measures, for example, absenteeism and presenteeism are measured at individual level by means of a survey. Productivity measures evaluated at the level of the organisation are evaluated over the period of 1 calendar year. The timeframe to evaluate absenteeism on individual level is 28 days, ie, participants are asked to indicate the number of scheduled work days they have missed during the past 28 days. The timeframe to evaluate presenteeism on individual level is 1 year.

### Data analysis

Regression models to estimate the association between the intervention and outcomes will include fixed effects of time, intervention and intervention by time. Hospitals represent higher order units of analysis, and clinicians the lower order unit of analysis. Countries will be treated as fixed effects. Repeated measures on nurses and physicians will also be appropriately modelled. More complex multilevel structural models will be employed to examine indirect and moderated associations between the process measures which we would expect to respond to the intervention and our outcome measures. All analyses will be conducted according to intention-to-treat. Data on hospital characteristics (eg, size, teaching status) and clinicians (eg, gender, age) will be collected and used to adjust for potential confounders in the regression analyses. Statistical significance will be assessed at the 5% level. Occasional missing data will be limited by the electronic survey implementation using a force response option for each item. Data analysis will be conducted using SAS software, V.9.4 (and subsequent releases) of the SAS system for windows (SAS Institute, Cary, North Carolina).

### Process evaluation
#### Qualitative data collection

Data will be collected on context, implementation and mechanisms of impact. Based on the results at baseline measurement, a purposive sample of two hospitals per country will be selected from the immediate intervention group for case studies to explore different aspects of the implementation process. Hospitals with the lowest burnout scores among clinicians will be contrasted with hospitals with the highest to explore potential differences in motivation and ambition in the hospitals. In total, four qualitative data collection periods are planned throughout the project (figure 1). Two focus group interviews (one with 6–8 nurses and one with 6–8 physicians) will be conducted in each of the two case hospitals (per country) at one point in time (M34-35). This will generate 20 focus groups in total across all countries. A purposive sampling approach is used to recruit participants. We will invite physicians and nurses who work in direct patient care in clinical areas targeted by the Magnet intervention and most closely involved in implementing change. By doing so, we will aim for diversity within each staff group in terms of clinical areas and length of clinical experience. If focus groups cannot be convened (eg, recruitment difficulties), then each focus group will be replaced with 2 to 3 one-to-one interviews with similar staff types to the target focus group sample. Depending on practicalities, this may result in a data set that is generated from focus groups and/or one-to-one interviews. A minimum of 40 and maximum of 160 individuals will be involved in this part of the study.

In addition to the focus groups, two semi-structured individual interviews will be conducted at three time points (months 23, 34 and 44) in each of the two case hospitals (per country). Persons directly involved in leading the implementation, that is, the Magnet4Europe intervention coordinator as well as another member of the interdisciplinary Magnet council will be invited to participate. This will lead to 60 interviews in total (across countries) over the course of the project. Where possible, we aim to interview the same two people across all time points. If individuals leave the study, we will aim to recruit a replacement with a similar role and professional background. We will also aim for maximum diversity in professional backgrounds, for example, one from nursing, one from medicine. A minimum of 30 and a maximum of 60 individuals will be involved in this part of the study.

To complement the qualitative evaluation, the 12-item Organisational Readiness for Implementing Change (ORIC) scale[95] and the 23-item Normalisation Measure Development Questionnaire (NoMAD)[96 97] will be used to explore the readiness for change and the extent to which Magnet is integrated and sustained in the hospital over time. This will be done after the start of the intervention and at various time points. Magnet4Europe intervention coordinators and all other members of the interdisciplinary Magnet4Europe council in all hospitals will be invited to respond to the surveys. A minimum of 54 and a maximum of 324 individuals across all countries will be involved in this part of the study at each time point. If individuals leave and are replaced, their replacements will be invited to complete the survey on the next occasion.

### Qualitative data analysis

Analysis of the qualitative data generated through focus groups and individual interviews will be guided by the

---

**Box 1  Magnet4Europe implementation evaluation**

**Context**
⇒ General information about the hospital (eg, type of hospital, size, organisational models, professional structure).
⇒ Hospital policy documents (eg, about staffing, work environment, leadership development, organisational plans).
⇒ Generalgeneral country information (eg, relevant legislation, financing of the healthcare sector).

**Implementation**
⇒ Results from the Magnet4Europe Magnet Gap Analysis Tool and the work plan of each hospital.
⇒ Implementation process survey data.
⇒ Data from interviews and focus groups.

**Mechanisms of impact**
⇒ Hospital policy documents (eg, staffing, work environment, leadership development, organisational plans).
⇒ Implementation process survey data.
⇒ Data from interviews and focus groups.

---

Framework Analysis approach.[98] Data collection and analysis will be conducted by researchers in each of the Magnet4Europe countries, using a common coding template, which will be developed based on initial themes. International meetings will be held regularly to discuss analysis and findings. NVivo V.12 (and subsequent releases) will be used for management and analysis of qualitative data.[99]

Data generated through collection of policy documents and general hospital information about context will be analysed through content analysis and/or discourse analysis[100] to explore different aspects of the organisational context in relation to the implementation process.

In the process evaluation, we will analyse how organisational context links to the impact and sustainability of the Magnet programme. Emerging themes from the qualitative analysis will be used, together with the results from descriptive analyses of our quantitative measurements of readiness for change and of implementation normalisation (NoMAD instrument), to explain variations in implementation between and within individual settings over time (box 1).

### Patient and public involvement

Patients and/or public were not involved in the design or development of research questions of this study.

### ETHICS AND DISSEMINATION
### Ethics

This study was approved by the Ethics Committee Research UZ/KU Leuven, Belgium; additionally, ethics approval is obtained in all other participating countries either through a central or decentral authority, to conduct nurse surveys as well as collect qualitative data. Any protocol amendments will be promptly reported to all relevant parties. An Ethical Advisory Committee was installed at the start of project and includes an independent ethics advisor with documented expertise to monitor ethics issues. The study is registered in the ISRCTN registry.

### Dissemination

Magnet4Europe has a strong focus on disseminating evidence on the impact of work environment on mental health of health professionals. Findings will be published in peer-reviewed journals and through other channels designed to reach a diverse community of researchers, practitioners and other stakeholders. Gold open access will be used for the key findings of the study to ensure maximum exposure to a practice community. Beyond the funded life of the project, the aim is to maximise accessibility by fully exploiting the opportunity for green open access and, where possible, identifying additional funds to support further gold open access publishing. Data management is compliant in line with the European Commission's Horizon 2020 Data Management Plan guidelines and FAIR-principles.

### DISCUSSION

This paper describes the theoretical framework, conceptual design and methodological approaches of the EU H2020 Magnet4Europe project, which aims to evaluate the transfer, implementation, scale up and cost-effectiveness of the Magnet model of organisational redesign in a European context as a system-level approach to improve clinician well-being. Mental health and well-being is among one of the highest priorities of the public health agenda in the EU.[1] This is based on the growing awareness and recognition of the magnitude of mental health problems and the burden on individuals, society and economy. With the extraordinary conditions created by the COVID-19 pandemic, this issue has taken an even more central role in the policy landscape.[20] COVID-19 is expected to have psychological impacts for health professional, in particular, frontline workers that will be sustained over time.[101–104]

The research activities conducted within the project are expected to improve significantly the work environment of health professionals in hospitals, especially after the COVID-19 pandemic. In particular, Magnet4Europe proposes an organisational intervention, targeting the root of many of the problems (ie, organisational and work-related stressors) facing health systems rather than focusing narrowly of their symptoms. It does so by means of a multicentre study involving more than 60 hospitals across six European countries. By choosing a cluster randomised controlled design, this study is the first with the power to identify a causal association between implementing healthy workplace characteristics and improved staff well-being in European hospitals on a large scale. Two international pilot initiatives testing the transferability of Magnet key principles and accompanied by rigorous evaluations have informed the design of Magnet4Europe.[46 47] In both, the intervention hospitals improved their work

environments substantially from their baselines at the end of the 2-year intervention, and outcomes for staff improved. The Magnet4Europe intervention builds on lessons learnt from these pilot initiatives. The degree of innovation can also be seen in the use of a multifaceted implementation strategy involving one-to-one twinning with international Magnetdesignated hospitals, creation of a critical mass, co-creation of solutions, development of learning collaboratives, benchmarking and interdisciplinary and management buy-in. In addition, Magnet4Europe makes a strong scientific contribution as it goes beyond traditional studies, which focus solely on individual professions and disciplines separately. This allows us to study the interaction between physicians and nurses working in teams and the impact of healthy environments on their well-being and patient outcomes.

Magnet4Europe will also allow countries to learn from the experience of other health systems and understand the factors influencing their sustainability, as the study will be undertaken in six European countries representing the two dominant health system forms. The so-called Beveridge countries (England, Ireland, Sweden, Norway) are tax funded resulting in potentially more government health policy decision-making while the so-called Bismarck countries (Belgium, Germany) have an employer-based social insurance financing system, which potentially allows for more organisational variation. This study allows us to compare and contrast the impact of healthcare systems on the implementation and sustainability of the intervention.

The project involves dissemination and stakeholder engagement. Various stakeholders will be involved in the project including designees from international and European organisations who represent nursing professionals, healthcare employers, patients and governments. Their main role is to raise awareness of the project and to support dissemination of the results as well as the Magnet4Europe consortium in formulating policy recommendations based on the scientific results.

## Limitations

Contextual factors inherent to the countries where Magnet4Europe will be implemented, more specifically different health systems within and across countries, may impact the potential for uptake of the intervention and unintentionally lead to variation in the intervention and outcomes. Certain healthcare system characteristics such as labour supply policies, education requirements and funding schemes may interact with the intervention. In predominantly Beveridge-type system countries, the interaction between these characteristics and the intervention may lead to a more cumbersome process of change and innovation compared with countries with a more decentralised insurance-based health system.

Despite having a strong conceptual design for the Magnet4Europe study, that is, a cluster randomised controlled trial, bearing many advantages and having the potential to facilitate the robust evaluation of this type

of intervention, this design is susceptible to multiple sources for potential risk of bias. First, contamination of the wait list control group cannot be ruled out. All hospitals were, prior to the start of the intervention, informed about the intervention and the active components. It was not possible to blind hospitals to their allocation in either the immediate intervention group or wait list control group. We aimed to mitigate contamination of hospitals assigned to the wait list control group by not providing them with the Magnet blueprint of organisational redesign and by not disclosing the name of their future Magnet twinning partner to avoid early contact and exchange of information prior to their actual start of the intervention. A second additional risk related to contamination is when the intervention takes longer to be fully embedded as intended. The intervention proposed in Magnet4Europe is a complex intervention at the hospital level, earlier research findings demonstrated that a considerable amount of time is required for the intervention to become fully embedded into practice and influence outcomes accordingly. Due to the limited time frame available in Magnet4Europe, no transition period is incorporated. This potential risk is mitigated by two intervention components, that is, the one-to-one twinning of the European intervention hospitals with a designated and experienced US Magnet hospital and the Learning Collaboratives. Both components are expected to act as a catalysator for knowledge transfer and innovation and expedite the implementation of the intervention. There is a potential risk that the true effect of the intervention only becomes apparent after the formal ending of the 4-year study period. A longer follow-up of the study beyond the funding is foreseen.

**Author affiliations**
[1]Department of Public Health and Primary Care, Katholieke Universiteit Leuven, Leuven, Flanders, Belgium
[2]Center for Health Outcomes and Policy Research, University of Pennsylvania, Philadelphia, Pennsylvania, USA
[3]Faculty of Health Sciences, University of Southampton, Southampton, UK
[4]Department of Healthcare Management, Technical University of Berlin, Berlin, Germany
[5]Occupational & Organisational Psychology and Professional Learning, Katholieke Universiteit Leuven, Leuven, Flanders, Belgium
[6]Optentia Research Unit, North-West University, Potchefstroom, South Africa
[7]School of Nursing and Midwifery, University College Cork, Cork, Ireland
[8]Department of Neurobiology, Care Sciences and Society, Karolinska Institutet, Stockholm, Sweden
[9]Department of Health Services Research and Policy, London School of Hygiene and Tropical Medicine, London, UK
[10]Florence Nightingale Faculty of Nursing, King's College London, London, UK
[11]Department of Psychology, Utrecht University, Utrecht, The Netherlands
[12]Department of Learning, Informatics, Management and Ethics, Karolinska Institutet, Stockholm, Sweden

**Acknowledgements** We thank the American Nurses Credentialing Center (ANCC) for granting the right to use Magnet® model as core element of the Magnet4Europe intervention. The Magnet® model is a trademark of ANCC registered in the USA and other jurisdictions and is being used under license from ANCC. All rights are

reserved by ANCC. ANCC's consent to the use of the Intervention Hospital Magnet® mark shall not be construed as ANCC sponsoring, participating, or endorsing the Magnet4Europe intervention. ©American Nurses Credentialing Center. Reproduced under license from the American Nurses Credentialing Center. All rights reserved.

**Collaborators** Magnet4Europe consortium: Walter Sermeus (director), Luk Bruyneel, Hans De Witte, Wilmar B. Schaufeli, Simon Dello, Dorothea Kohnen (Belgium, Catholic University Leuven); Linda Aiken (codirector), Matthew McHugh, Mary Del Guidice, Herbert Smith, Timothy Cheney, Douglas Sloane (USA, University of Pennsylvania); Reinhard Busse, Claudia Maier, Julia Köppen, Joan Kleine (Germany, Technical University Berlin); Jonathan Drennan, Vera McCarthy, Elaine Lehane, Noeleen Brady, Anne Scott (Ireland, University College Cork); Ingeborg Strømseng Sjetne (Norway, Norwegian Institute of Public Health); Anners Lerdal, Monica Bukkøy Kjetland (Norway, Lovisenberg Diaconal Hospital); Lars E. Eriksson, Rikard Lindqvist, Lisa Smeds Alenius (Sweden, Karolinska Institutet); Jane Ball, Peter Griffiths, Jackie Bridges, Sydney Anstee (England, University of Southampton); Anne Marie Rafferty (England, King's College London); Martin McKee (England, London School of Hygiene and Tropical Medicine); Paul Van Aken, Danny Van Heusden, Kaat Siebens, Peter Van Bogaert (Belgium, University Hospital Antwerp).

**Contributors** WS, LHA, MDM and LB devised the study. WS, LHA, MDM and LB drafted the protocol, with assistance from JBa, JBr, RB, HDW, SD, JD, LEE, PG, DK, JK, RL, CBM, MM, AMR, WBS, DMS, LSA, HS. All authors provided edits and reviewed the manuscript for important intellectual content. All authors gave final approval of the version to be published and agreed to be accountable for all aspects of this work.

**Funding** This study is funded by the European Union's Horizon 2020 research and innovation programme under the project Magnet4Europe: Improving Mental Health and Wellbeing in the Health Care Workplace (Grant Agreement Number 848031), starting 1 January 2020 until 31 December 2023. The materials presented here are the responsibility of the authors only. The EU Commission takes no responsibility for any use made of the information set out.

**Competing interests** None declared.

**Patient and public involvement** Patients and/or the public were not involved in the design, or conduct, or reporting, or dissemination plans of this research.

**Patient consent for publication** Not applicable.

**Provenance and peer review** Not commissioned; externally peer reviewed.

**ORCID iDs**
Walter Sermeus http://orcid.org/0000-0002-5915-1845
Jackie Bridges http://orcid.org/0000-0001-6776-736X
Simon Dello http://orcid.org/0000-0001-6043-1206
Lars E Eriksson http://orcid.org/0000-0001-5121-5325
Peter Griffiths http://orcid.org/0000-0003-2439-2857
Dorothea Kohnen http://orcid.org/0000-0002-7220-1819
Matthew D McHugh http://orcid.org/0000-0002-1263-0697
Martin McKee http://orcid.org/0000-0002-0121-9683

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
