## [Reviewer comments · BMJ Open]

ARTICLE DETAILS

TITLE (PROVISIONAL)	A workplace organizational intervention to improve hospital nurses' and physicians' mental health: study protocol for the Magnet4Europe wait-list cluster randomized controlled trial
AUTHORS	Sermeus, Walter; Aiken, Linda; Ball, Jane; Bridges, Jackie; Bruyneel, Luk; Busse, Reinhard; De Witte, Hans; Dello, Simon; Drennan, Jonathan; Eriksson, Lars E; Griffiths, Peter; Kohnen, Dorothea; Köppen, Julia; Lindqvist, Rikard; Maier, Claudia; McHugh, Matthew; McKee, Martin; Rafferty, Anne Marie; Schaufeli, Wilmar; Sloane, Douglas; Alenius, Lisa Smeds; Smith, Herbert; Magnet4Europe consortium, Magnet4Europe consortium

VERSION 1 – REVIEW

REVIEWER	Herr, Raphael Heidelberg University, Mannheim Institute of Public Health, Social and Preventive Medicine
REVIEW RETURNED	17-Jan-2022

GENERAL COMMENTS	This is the study protocol of a very interesting and relevant study. In my opinion, all relevant aspects are adequately described. I wish the authors much success with the implementation!
---

REVIEWER	Barchielli, Chiara Scuola Superiore Sant'Anna, Nursing
REVIEW RETURNED	14-Feb-2022

GENERAL COMMENTS	I was really impressed by Your work. The paper addresses pivotal issues and sheds light on one above all, the mental distress of healthcare worker, which seems to be heavily underestimated. The RCT and the methodology followed allow to have robust results. I am living in a country in which the Magnet programme is not active, and after reading your work, I wish even more that it was running. I strongly think Your work is outstanding and of course must be published.
--

REVIEWER	Kaiseler, Mariana Leeds Beckett University Institute for Sport Physical Activity and Leisure, School of Sport
REVIEW RETURNED	28-Feb-2022

GENERAL COMMENTS	Timely project with a comprehensive inclusion criteria and novel research approach. Some areas for development include explaining the intervention Magnet for Europe further in the introduction (i.e., levels of influence; why is it different from other existent organisational interventions?) Methodology is carefully explained, however there are some
--

	aspects that deserve further consideration:  - why is the programme subject to travel/ can intervention not be delivered remotely? - how will qualitative data be integrated with quantitative data? - what is the rationale for 8 weeks intervention? - what is the productivity measures timeframe (i.e., retention for how long?) and why? - diversity is mentioned but not explained (how will diversity be considered? Gender/ethnic/minority groups?) The project keeps mentioning hospitals in Europe but England is not part of Europe, so please review and adapt. This has been done in certain parts of the paper but not consistently throughout. Finally, consider a timeframe for development and implementation so the audience can be aware of when this will take place.
--	--

VERSION 1 – AUTHOR RESPONSE

Comments to the author from Dr. Marian Kaiseler (reviewer 3)

Timely project with comprehensive inclusion criteria and a novel research approach. Some areas for development include:

1. *Explaining the intervention Magnet for Europe further in the introduction (i.e., levels of influence; why is it different from other existent organisational interventions?).*

Author reply: In the introduction of the manuscript, the Magnet4Europe intervention is now described and briefly explained. We also explain how the Magnet4Europe intervention differentiates itself from other organisational interventions. This point has been addressed in the manuscript in the following way on lines 122-133: “The aim of the Magnet4Europe project is to evaluate the ability of organizational redesign in European hospitals to improve nurses’ and physicians’ mental health and well-being. The organizational redesign intended in Magnet4Europe is a system level intervention, targeting the hospital as a whole on the level of the organization. Through co-creation and stakeholder co-designed adaptations, the Magnet® model is adapted to the European context, providing the first element and basis for the Magnet4Europe intervention. In addition to this first component, one-to-one twinning with Magnet® recognized hospitals, learning collaboratives and a critical mass of hospitals are also introduced into the intervention. While the Magnet4Europe intervention shares common elements with other organizational interventions, the combination of all the above components, targeting two major clinical professions – both nurses and physicians – and its primary focus on the work environment as levers for change differentiate the Magnet4Europe intervention from any other organizational intervention and make it unique in its kind.[48]”.

Methodology is carefully explained, however, there are some aspects that deserve further consideration:

2. *Why is the programme subject to travel/ can intervention not be delivered remotely?*

Author reply: The intervention employed in Magnet4Europe can be categorised as a complex intervention and entails multiple components. In the manuscript we first describe

how this specific component of the intervention was planned and intended. The paragraph below provides more information on why meeting in-person is important.

The aim of the one-to-one twinning and the Learning Collaboratives is to maximise efficiency, reduce time to successful implementation, and maximise the sharing of knowledge and information. When describing and explaining the effect of this component, we also cite the publication of Nembhard, I. (2009). Principal findings of this study highlight that - amongst other features – Learning Sessions interactions were found to be most helpful for advancing improvement efforts and knowledge acquisition. Learning Sessions are described as: “Teams attend a series of multiday meetings, known as “Learning Sessions”, where they learn improvement techniques from experts and share their experiences implementing new practices with one another”. The one-to-one twinning follows the principles described above, i.e. persons leading the intervention in the European intervention hospitals meet on multiple occasions throughout the project with their U.S. Magnet® twinning partner (i.e. expert in this case) to obtain knowledge and learn from the expert. In between the meetings, the intervention hospital implements and uses the new knowledge, facilitating the intended organisational redesign. Attending Learning Sessions implies that people meet one another “in person” and not virtually.

The COVID-19 pandemic was a disruptive event, forcing us to temporarily continue this aspect of the intervention an all-virtual manner. Organising virtual meetings, was the way we have managed to operationalise this part of the Magnet4Europe intervention during the COVID-19 pandemic.

In theory, it would not be impossible to deliver the intervention virtually, but we firmly believe that in order to deliver the intervention to its full potential and maximise the impact it is imperative that intervention hospitals meet in person with their twinning partner. To reflect this, we have mentioned that these types of meetings will be resumed as soon as travel is safe. This point has been addressed in the manuscript in the following way on lines 228-230: “Face-to-face meetings between both parties allow for a profound, impact- and meaningful experience that maximises the effect of the capacity building activities facilitating the intended organisational redesign.”.

3. How will qualitative data be integrated with quantitative data?

Author reply: Various types of quantitative data will be collected in Magnet4Europe. First, an extensive survey targeted at nurses and physicians providing direct patient care is used to measure a core set of outcomes and main explanatory and process variables directly linked to the Magnet4Europe intervention; allowing us to assess change over time. Secondly, within the process evaluation, the ORIC and NoMAD instrument is used to measure the organisational readiness for change and normalisation of the intervention. Within the process evaluation, also qualitative data is being collected. Quantitative data (i.e. level burnout on hospital level) is utilised to select case hospitals for the qualitative evaluation. This is described on lines 342-346. When we perform the process evaluation, qualitative data will be triangulated with the quantitative data providing a rich and comprehensive data set. This data encompasses information on various facets of the intervention and allows for a well-balanced and comprehensive evaluation of the implementation of the intervention. This is described in the manuscript on lines 386-390. No changes were made to the manuscript to address this point.

4. What is the rationale for 8 weeks intervention?

Author reply: The Magnet4Europe intervention is not limited to a period of 8 weeks. The Magnet4Europe intervention in the immediate intervention group started in month 10 of the

project (October 2020) and will have a total duration of 31 months. The intervention in the usual-practice wait-list control group is initiated in month 17 (May 2021) and will last for 24 months. A visual representation of the duration of the intervention is provided in figure 1 on line 198. We have reviewed the text and are unsure why reviewer 3 thought the intervention was 8 weeks. If the reviewer would care to highlight any misleading or confusing passages we will gladly amend. No changes were made to the manuscript to address this point.

5. *What is the productivity measures timeframe (i.e., retention for how long?) and why?*

Author reply: The productivity measures we aim to evaluate in Magnet4Europe at the level of the organisation are staff turnover rate, number of vacancies and the use of agency staff. On an individual level, we measure and evaluate absenteeism and presenteeism. The time frame used to evaluate the productivity measures is one calendar year. For example, to calculate and evaluate staff turnover at the level of the organisation, we ask each organisation for the number of employees (i.e. physicians and registered nurses more specifically) at the start of the year (e.g. 01.01.2019) and the number of employees who were continuously employed from the start of that year (01.01.2019) to the end (31.12.2019) (i.e. not counted those who left and those who were newly hired during that particular year). We addressed this point in the manuscript in the following way on lines 322-326: “secondary outcome measures, e.g., absenteeism and presenteeism are measured at the individual level utilizing a survey. Productivity measures evaluated at the level of the organisation are evaluated over the period of one calendar year. The timeframe to evaluate absenteeism on an individual level is 28 days, participants are asked to indicate the number of scheduled work days they have missed during the past 28 days. The timeframe to evaluate presenteeism on an individual level is one year.”

6. *Diversity is mentioned but not explained (how will diversity be considered? Gender/ethnic/minority groups?)*

Author reply: Diversity in the context where we describe it in this manuscript, i.e. in the qualitative data collection as part of the process evaluation, refers to diversity in the type of health professionals we aim to include in our sample of interviewees for both the focus groups and the semi-structured interviews. For the focus groups for example we aim to include both nurses and physicians, who have different specialities and variations in clinical experience. The aim of selecting a diverse sample is to obtain rich and saturated data from both the focus groups and semi-structured interviews. This is described on lines 350-353 and 364-365. No changes were made to the manuscript to address this point.

7. *The project keeps mentioning hospitals in Europe but England is not part of Europe, so please review and adapt. This has been done in certain parts of the paper but not consistently throughout.*

Author reply: In addressing this point, we feel it's important to make a clear distinction between the continent of Europe on the one hand and the European Union – being the most important partnership in Europe - on the other hand. England is not one of the 27 member states of the European Union, but is part of Europe, as is Norway. In the manuscript we have addressed this point in the following way on lines 139-141: “The setting will be acute general hospitals in six European countries, including both European Union member states (Belgium, Germany, Ireland and Sweden) and non-European member states (England and Norway).”

8. *Finally, consider a timeframe for development and implementation so the audience can be aware of when this will take place.*

Author reply: The timeframe for the development of the intervention is beyond the scope of this protocol. The development of the intervention was established during the grant proposal phase. The implementation of the intervention is currently ongoing. The timeframe of the intervention, including the start and end date, is depicted in figure 1 on line 198. To address this point and to further emphasise when the implementation is actually taking place, the year has been added to the very first row of figure 1 on line 198.